# Pyranometer offsets triggered by ambient meteorology: insights from laboratory and field experiments

**Sandro M. Oswald**[1, 2, 3]**, Helga Pietsch**[2]**, Dietmar J. Baumgartner**[4]**, Philipp Weihs**[3]**, and Harald E. Rieder**[1, 2, 5]

[1]Wegener Center for Climate and Global Change, Graz, Austria
[2]Institute for Geophysics, Astrophysics and Meteorology/Institute of Physics, Graz, Austria
[3]Institute of Meteorology, University of Natural Resources and Applied Sciences (BOKU), Vienna, Austria
[4]Kanzelhöhe Observatory for Solar and Environmental Research, Graz, Austria
[5]Austrian Polar Research Institute, Vienna, Austria

*Correspondence to:* Sandro M. Oswald (sandro.oswald@boku.ac.at)

**Abstract.** This study investigates effects of ambient meteorology on the accuracy of radiation measurements performed with pyranometers contained in various heating/ventilation systems (HV-systems). It focuses particularly on instrument offsets observed following precipitation events. To quantify pyranometer responses to precipitation, a series of controlled laboratory experiments as well as two targeted field campaigns were performed in 2016. The results indicate that precipitation (as simulated by spray-tests or observed under ambient conditions) significantly affects the thermal environment of the instruments and thus their stability. Statistical analysis of laboratory experiments showed that precipitation triggers zero offsets of $-4\,\mathrm{W\,m^{-2}}$ or more, independent of the HV-system. Similar offsets have been observed in field experiments under ambient environmental conditions, indicating a clear exceedance of BSRN targets following precipitation events. All pyranometers required substantial time to return to their initial signal states after the simulated precipitation events. Therefore for BSRN class measurements the recommendation would be to flag the radiation measurements during a natural precipitation event and $90\,\mathrm{min}$ after it in nighttime conditions. Further daytime experiments show pyranometer offsets of $50\,\mathrm{W\,m^{-2}}$ or more in comparison to the reference system. As they show a substantially faster recovery, the recommendation would be to flag the radiation measurements within a natural precipitation event and $10\,\mathrm{min}$ after it in daytime conditions.

## 1 Introduction

Earth's climate is largely determined by the global energy balance (Wild et al., 2012). Therefore a precise knowledge of the surface energy budget, which includes the solar and terrestrial radiation fluxes, is essential for understanding the Earth's planetary circulation and climate system (Ramanathan, 1987; Augustine and Dutton, 2013; Wild et al., 2014).

In situ measurements of solar radiation on the Earth's surface, more precisely global radiation which is the sum of the direct and diffuse components, began in the 1920s, but became more widespread with the advent of thermopile pyranometers and through initiatives of the International Geophysical Year 1957/58 (Wild, 2009). Around the turn of the century a series of studies (Dutton et al., 1991; Gilgen et al., 1998; Ohmura et al., 1998; Stanhill, 2005; Liepert, 2002) reported negative trends of global radiation based on in-situ measurements, a phenomenon commonly referred to as ‚global dimming‘ (Wild, 2005, 2009). Average trends of $-6$ to $-9\,\mathrm{W\,m^{-2}}$ between 1960-1990 have been reported in the literature (Wild, 2005), but estimates vary depending on location, record length and time period considered (Wild et al., 2012). The previously observed negative trends have been replaced by a widespread increase in surface solar radiation over the period 1990-2000, a phenomenon commonly referred to as ‚global brightening‘ (Wild, 2005).

The growing interest of the scientific community in surface radiation trends and limitations in the accuracy of historic records led in the early 1990s to the establishment of the Baseline Surface Radiation Network (BSRN) under the auspices of the World Climate Research Programme (WCRP)

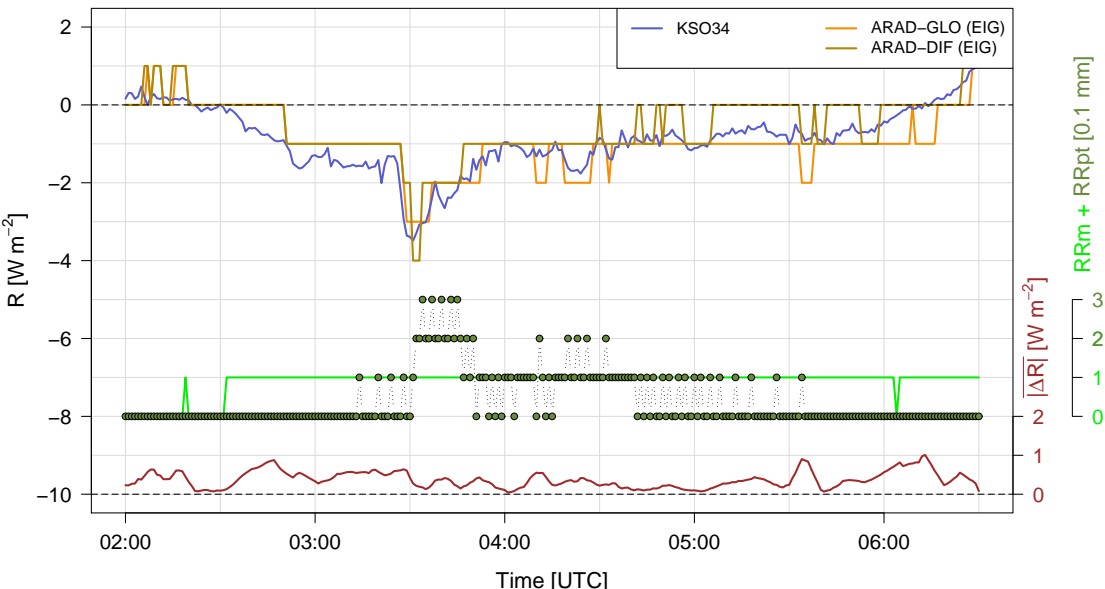

**Figure 1.** Natural event of a simultaneous decrease of radiation (R) measured with three CMP21 pyranometers in various heating/ventilation systems at ARAD site Graz/University on the 10th of February 2016. System acronyms represent measurements of global (ARAD-GLO) and diffuse (ARAD-DIF) solar radiation at the ARAD-platform (with CMP21 pyranometers contained in Eigenbrodt SBL 480 (EIG) HV-systems); global radiation measurements (KSO34) with an additional CMP21 pyranometer (contained in a KSO34 HV-system). Output of the precipitation sensor (RRm), and ombrometer (RRpt) operated at the co-located meteorological station Graz/University is shown along with the radiation measurements. Heavy precipitation started around 03:30 UTC.

(Ohmura et al., 1998). BSRN sites have been equipped with instruments of highest accuracy and to date more than 50 anchor sites are operational around the globe. Besides BSRN a series of national monitoring networks was established at this time operating at (or close) to BSRN standards.

One of these national monitoring networks is the so-called Austrian RADiation Monitoring Network (ARAD), which has been established in 2010 by a consortium of the Central Agency of Meteorology and Geodynamics (ZAMG), the University of Graz, the University of Innsbruck, and the University of Natural Resources and Applied Sciences, Vienna (BOKU). ARAD aims to provide long-term monitoring of radiation budget components at highest accuracy and to capture the spatial patterns of radiation climate in Austria (Olefs et al., 2016). To date the ARAD Network comprises one BSRN site (Sonnblick) and five additional sites (Kanzelhöhe Observatory, Graz/University, Innsbruck, Klagenfurt and Wien Hohe Warte). All ARAD sites are equipped with instrumentation according to BSRN standards (McArthur, 2005).

Despite BSRN class equipment and regular instrument maintenance, radiation measurements are also influenced by meteorological conditions and instrumentation effects leading occasionally to so-called zero offsets (Kipp and Zonen, 2010).

Field measurements performed within the scope of ARAD indicate that such zero offsets are frequently triggered by precipitation events. However, to the knowledge of the authors

to date no study has systematically investigated the influence of precipitation events on the accuracy of radiation measurements.

This study aims to close this gap by investigating the influence of precipitation events on the accuracy of radiation measurements under laboratory and field conditions. Three measurement campaigns, one under controlled laboratory conditions and two under ambient environmental conditions, have been performed between January and May 2016. The campaign design was centered on zero offsets during nighttime conditions and on the influence of precipitation events on the accuracy of radiation measurements in the ARAD setup.

The particular interest in the influence of precipitation events stems from the regular observation of zero offsets (during nighttime conditions) following precipitation events within the ARAD network. Figure 1 illustrates such an event in the series of global and diffuse radiation (R) measurements at ARAD site Graz/University.

## 2   Methods and Instrumentation

During all campaigns radiation measurements have been performed with a series of pyranometers of type CMP21 (manufactured by Kipp&Zonen) which are routinely operated for the measurement of global (GLO) and diffuse (DIF) solar radiation at the majority of ARAD sites. The CMP21 pyranometer is composed of two quartz glass-domes, a black

**Table 1.** Characteristics of the different heating/ventilation systems used in this study.

|  | System I | System II | System III |
|---|---|---|---|
| Manufacturer | Eigenbrodt GmbH & Co. KG | Observatory Kanzelhöhe | PMOD[1], World Radiation Center |
| Type | SBL 480 | KSO34 | PMOD-VHS |
| Acronym | EIG | KSO34 | DAV |
| Power | 24 VAC | 24 VAC | 24 VAC[2] |
| Ventilation | continuously | continuously | continuously |
| Heating power | 10 W | 10 W | 10 W |
| Heating element | discrete electrical resistor | discrete electrical resistor | circular heating element |

[1]Physikalisch-Meteorologisches Observatorium Davos. [2]modified by ZAMG for use with 24 VAC, original PMOD config. is for use with 12 VDC.

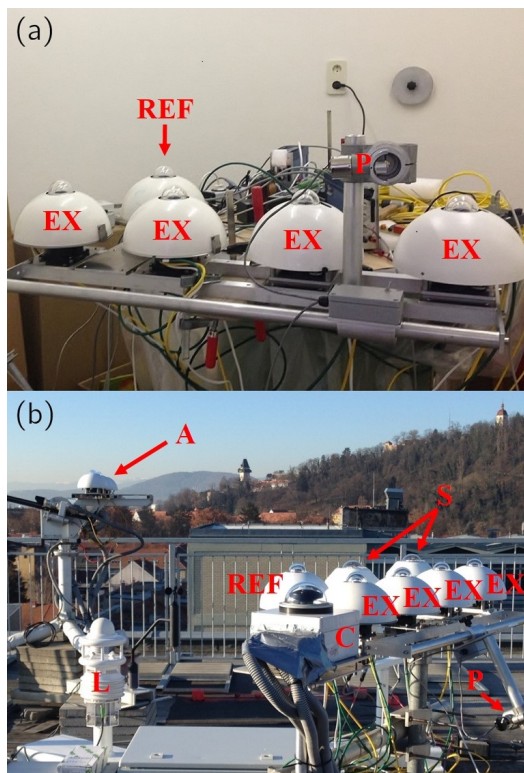

**Figure 2.** Measurement setup at (a) Kanzelhöhe Observatory and (b) the measurement platform of Graz/University in direct vicinity of the ARAD site. During field campaigns the measurement setup was expanded by an ‚all-in-one' meteorological observing system for the monitoring of ambient meteorological conditions, two star pyranometers and a cloudcam. Acronyms indicate: (REF) the reference CMP21 pyranometer contained in an Eigenbrodt SBL 480 HV-system; (EX) the ‚experimental' CMP21 pyranometers contained in an Eigenbrodt SBL 480 HV-system, DAVOS-PMOD/WRC HV-system, self-built KSO HV-system KSO34 and a further self-built KSO HV-system; (P) the electric motor pump used for automated spray-tests; (L) the ‚all-in-one' meteorological observing system (WS600 UMB manufactured by Lufft GmbH); (C) the cloudcam (VIS-J1006, manufactured by CMS Schreder Gmbh); (S) two star pyranometers (type 8102, manufactured by Schenk); and (A) ARAD site Graz/University.

receiving area (sensing element), a thermal battery (thermopile), a thermalisation resistance/compensation element in the body, a thermistor for body temperature, and a drying cartridge. The black receiving area bases on a passive sensing element called thermopile, which consists of 16 thermocouple junction pairs connected electrically in series. The temperature of one of these thermocouple junctions, called active or ‚hot' junction, increases with the absorption of solar radiation. A reference or ‚cold' junction, fixed on the thermopile, is held at a constant temperature and serves as reference for the ‚hot' junction. The differential temperature between the ‚hot' and ‚cold' junction produces an electromotive force directly proportional to the difference in temperature and is converted to an output voltage corresponding to the absorbed solar radiation. This process is referred to as Seebeck effect. As every thermal battery has its own physical properties and structure, every radiometer has its specific and individual calibration factor. The black receiving area has a very rough surface structure with micro-cavities that effectively absorbs more than 97% of the short-wave radiation in a broad spectral range from 300 to 3000 nm. CMP21 pyranometers are complying with the ISO 9060 standard and the guidelines of the World Meteorological Organization (WMO) (Kipp and Zonen, 2010).

The body temperature of a pyranometer of the CMP series is measured by a thermistor (type: YSI-44031, 10 kΩ @ 25 °C). This body temperature is directly proportional to the ambient air temperature whereby the possibility of the emergence of heat currents in the radiometer, causing a so-called *Zero Offset type B*, has to be considered. Such zero offsets are specified by the manufacturer, to occur following a $5\,\mathrm{K\,h^{-1}}$ change in ambient air temperature (Kipp and Zonen, 2010) over short time intervals.

Pyranometers used within the ARAD network are operated in different heating and ventilation systems (hereinafter referred to as HV-systems). Thus the entire set of ARAD HV-systems, comprising the commercially available Eigenbrodt SBL 480 (hereinafter referred to as EIG) and DAVOS-PMOD/WRC (hereinafter referred to as DAV). In addition to those a self-built HV-system manufactured by the staff of Kanzelhöhe Observatory (KSO34), have been used during

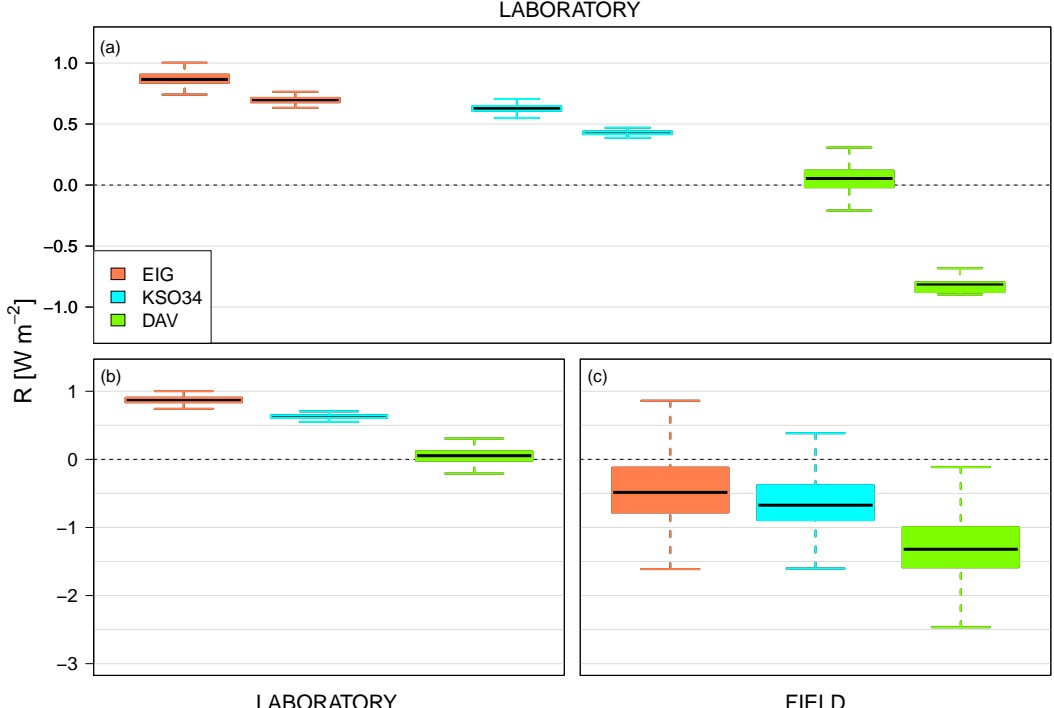

**Figure 3.** Spread of radiation measurements (R) with CMP21 pyranometers contained in different HV-systems (EIG, KSO34 and DAV) during dormant (i.e., undisturbed) phases during (a) and (b) laboratory and (c) field campaigns. Panel (a) provides results for two CMP21 pyranometers per HV-system as multiple instruments have been available during the laboratory campaign.

all campaigns. An overview about the characteristics of individual HV-systems is provided in Table 1. Serial numbers of HV-systems and CMP21 pyranometers are provided for completeness in Table S1 in the supplemental material to this article.

In addition to radiation measurements, standard meteorological observations of temperature, precipitation, relative humidity of air and wind speed and direction have been performed during campaigns.

As our investigations focused mainly on the question how precipitation events influence the accuracy of radiation measurements, a series of spray-tests has been performed during all campaigns. These spray-tests have been performed with an automated pumping system (designed and built by the staff of Kanzelhöhe Observatory), applying 30 strokes of distilled water (approx. $3.4$ ml) from a vertical distance of 6 cm on to the pyranometer's glass dome. The spray system created very fine, homogeneous drizzle, producing small droplets on the pyranometer dome, which quickly coagulated to larger drops. Such coagulation on pyranometer domes is also observed during stratiform and convective precipitation events.

CMP21 pyranometers have been operated, in parallel, in different HV-systems during a laboratory campaign at Kanzelhöhe Observatory (KSO, in January 2016) and during two field campaigns (one each in January and April/May

2016) at the measurement platform of the University of Graz in direct vicinity of the ARAD site (see Fig. 2).

During the measurement campaigns all CMP21 pyranometers have been operated in the standard ARAD configuration for global radiation measurements at low- to mid-altitude sites (heating level 10 W).

All measurement systems (i.e., pyranometers and HV-systems) were mounted in series on a stable aluminum jig, and a slide bar on the jig ensured seamless position changes of the electric motor pump for spray-tests.

The first measurement campaign was performed between 6th and 17th January 2016 in the laboratory of Kanzelhöhe Observatory. During this campaign all pyranometers/HV-systems have been operated under controlled ambient conditions at an approximately constant air temperature of $T_a \approx 7$ °C and approximately constant relative humidity of RH $\approx$ 65%. As we are particularly interested in zero offsets, pyranometers have been operated in the dark. Figure 2a provides an overview about the measurement setup in the laboratory of Kanzelhöhe Observatory.

Following the laboratory experiments, two field campaigns (18th to 25th January 2016 and 12th April to 3rd May 2016) have been performed. Figure 2b shows the measurement setup during field campaigns in direct vicinity of the ARAD site Graz/University.

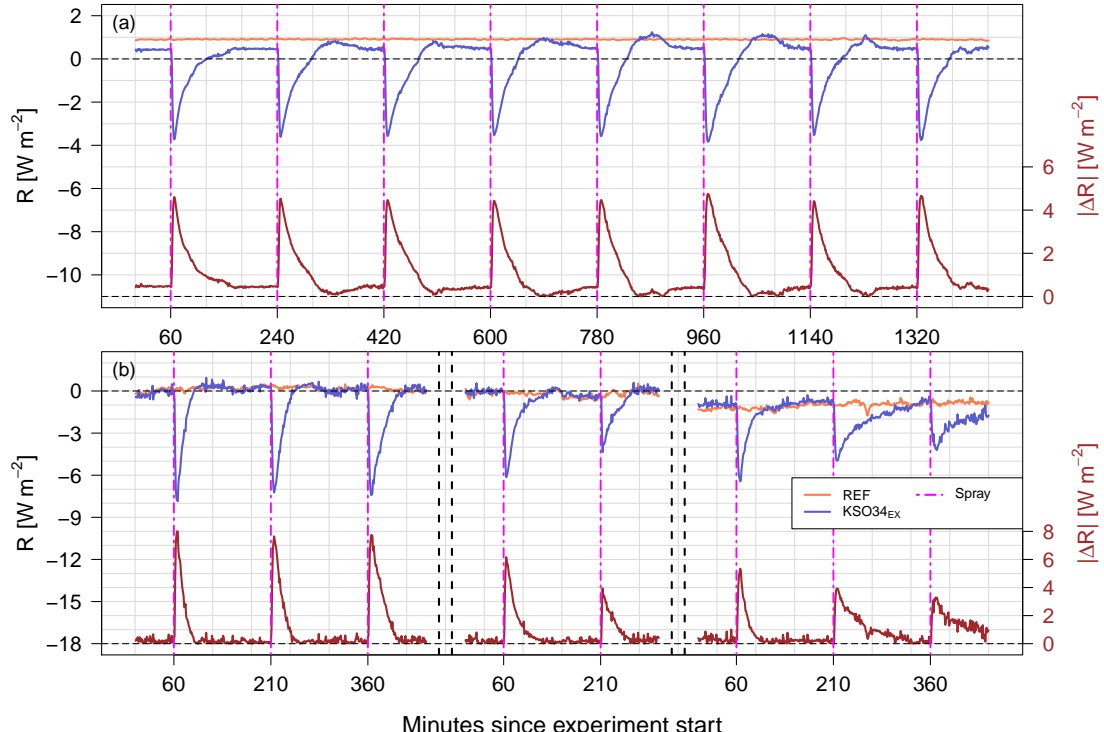

**Figure 4.** Time series of the radiation (R) measured by the reference (REF, coral, Eigenbrodt HV-system) and experimental pyranometer (KSO34$_{EX}$, blue, KSO HV-system) and absolute difference ($|\Delta R|$) between REF and KSO34$_{EX}$ during (a) laboratory conditions and (b) ambient environmental conditions. All field measurements have been performed during nighttime. Measurement series is continuous in (a) while start point of subpanels (separated by vertical double dashed lines) in (b) is always 18:30 UTC. Note: scales of y-axes differ between panels.

During laboratory and field campaigns for each pyranometer/HV-system combination a series of eight controlled spray-tests has been performed.

## 3 Results

### 3.1 Laboratory experiments

#### 3.1.1 Comparison of pyranometers under undisturbed conditions

Initial investigations of the laboratory campaign have been centered on the comparison of CMP21 performance, when operated within the different HV-systems used within the ARAD network. To this aim, pyranometer output was compared during dormant phases (without external impact factors). This comparison focused on (i) the temporal stability of pyranometer/HV-system combinations when operated in a steady environment and (ii) differences in the absolute values of the pyranometer outputs when operated in different HV-systems under the same controlled ambient conditions. Differences and spread in pyranometer output values have been established over a measurement interval of 65 h, following

a 24 h system spin up phase, and are summarized in Fig. 3. Under controlled laboratory conditions differences among CMP21 pyranometers operated in the same HV-system have been on average smaller than 1 W m$^{-2}$ and output values of CMP21 pyranomters across HV-systems have been within $\pm 1$ W m$^{-2}$.

Given the general stable performance of pyranometers within each HV-system and the small differences in output values (we note that most of the ARAD sites resolve pyranometer output at coarser resolution than during experiments, i.e. 1 W m$^{-2}$ steps) an Eigenbrodt SBL 480 HV-system has been used as housing for the undisturbed reference pyranometer (REF) during all experiments (laboratory and field), as it is also the most frequently used HV-system within the ARAD network (Olefs et al., 2016).

### 3.1.2 Spray-tests under controlled conditions

After the initial instrument comparison a series of spray-tests was performed for each pyranometer/HV-system combination.

Experiments comprised eight spray-tests per pyranometer/HV-system combination, each with 30

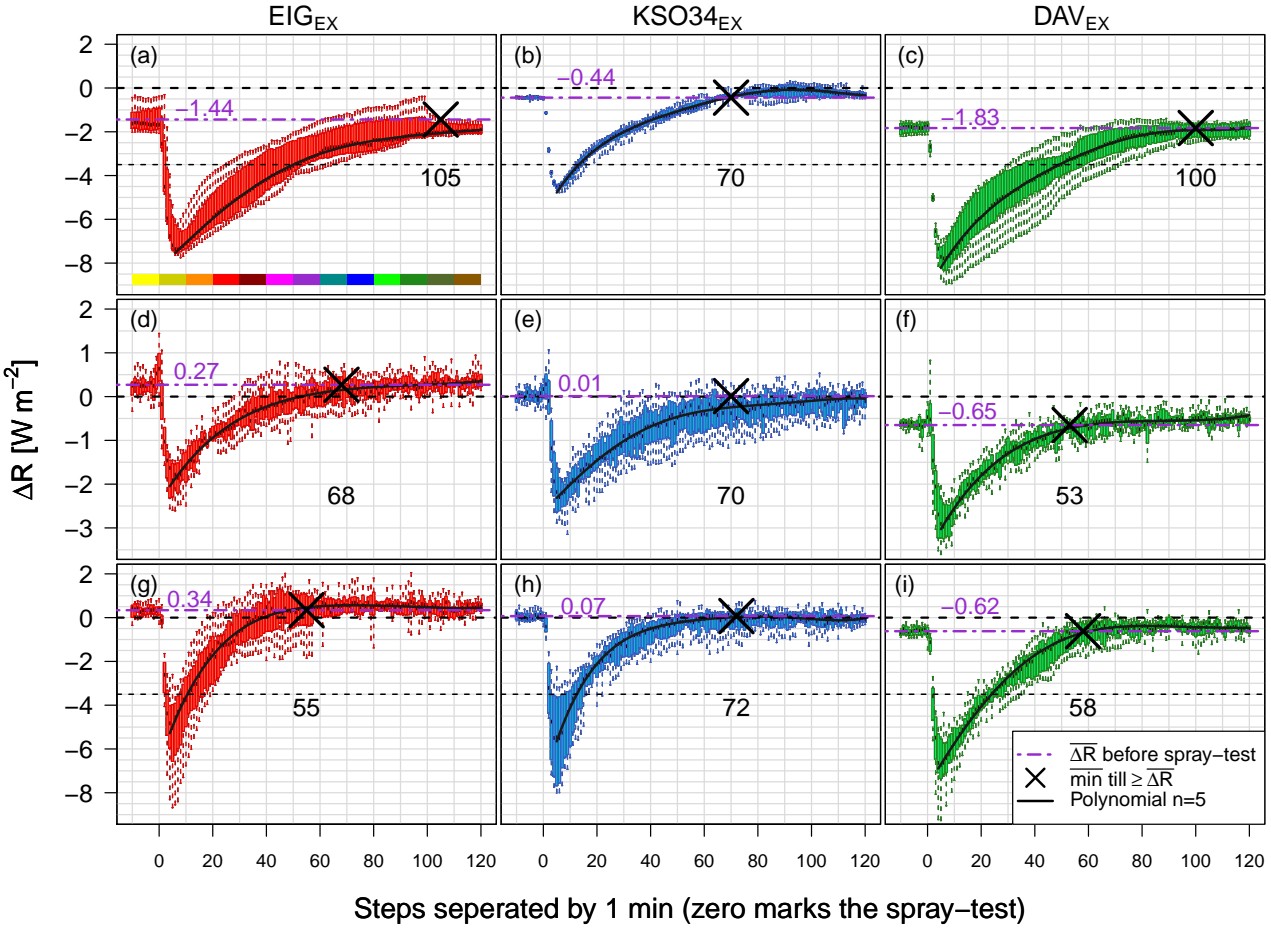

**Figure 5.** Box-whisker plots (in one minute time intervals) of the difference in radiation measurements ($\Delta R$) between individual experimental pyranometer/HV-system combinations (EIG$_{EX}$ in red, KSO34$_{EX}$ in blue and DAV$_{EX}$ in green) and the reference pyranometer (REF) following spray-tests. Subpanels (a)-(c) show results from experiments during the laboratory campaign, subpanels (d)-(f) show results from the first field campaign (January 2016) and subpanels (g)-(i) of the second field campaign (April/May 2016), respectively. The x-axis in all panels shows experiment time, starting 10 min before and ending 120 min after the spray-tests (marked with zero). The purple dotdashed horizontal line marks the average difference $\overline{\Delta R}$ before spray-test. The black cross marks the average time in minutes which each pyranometer/HV-system pair needed to return to/exceed its initial state (numbers give corresponding time in minutes). The color bar in subpanel (a) represents the temporal evolution of experiments further analyzed in Fig. 7. In all panels a polynomial of 5th degree (black solid line) of the median values, beginning at the minimum (maximum pyranometer response), is shown. Note: scales of y-axes in (d)-(f) differ from those in (a)-(c) and (g)-(i).

strokes and a 3 h dormant phase between individual spray-tests, allowing for systems to recover to initial state. The amount of water applied per spray-test corresponded to approximately 3.4 ml. In the following, the experimental pyranometer/HV-system combinations are referred to as EIG$_{EX}$, KSO34$_{EX}$ and DAV$_{EX}$, respectively.

Figure 4a provides a time series of one of the spray-tests performed during the laboratory campaign. The output signal of KSO34$_{EX}$ decreased by approximately the same value ($\sim -4$ W m$^{-2}$) during each experiment and needed about $1-2$ h to recover to its initial state thereafter. Similar results have been obtained for other pyranometer/HV-system combinations (see Fig. S1a for EIG$_{EX}$ and Fig. S2a for DAV$_{EX}$ in the supplemental material to this article). The pyranometer

response to spray-tests is attributed to a change in the thermal balance, i.e., the outer glass dome experiences evaporative cooling during/following the spray-test.

The subpanels (a)-(c) in Fig. 5 show box-whisker plots of average pyranometer responses for the period spanning 10 min before to 120 min after the spray-tests (marked with zero) for EIG$_{EX}$, KSO34$_{EX}$ and DAV$_{EX}$. The purple dotdashed horizontal line marks the average difference $\overline{\Delta R}$ before spray-test, the black cross marks the average time in minutes which each pyranometer/HV-system needed to return to/exceed its initial state. Zero offsets exceeded $-4$ W m$^{-2}$ for all pyranometer/HV-system combinations, and offsets as large as $\sim -8.5$ W m$^{-2}$ have been observed. The recovery time to the initial state following spray-tests ranged

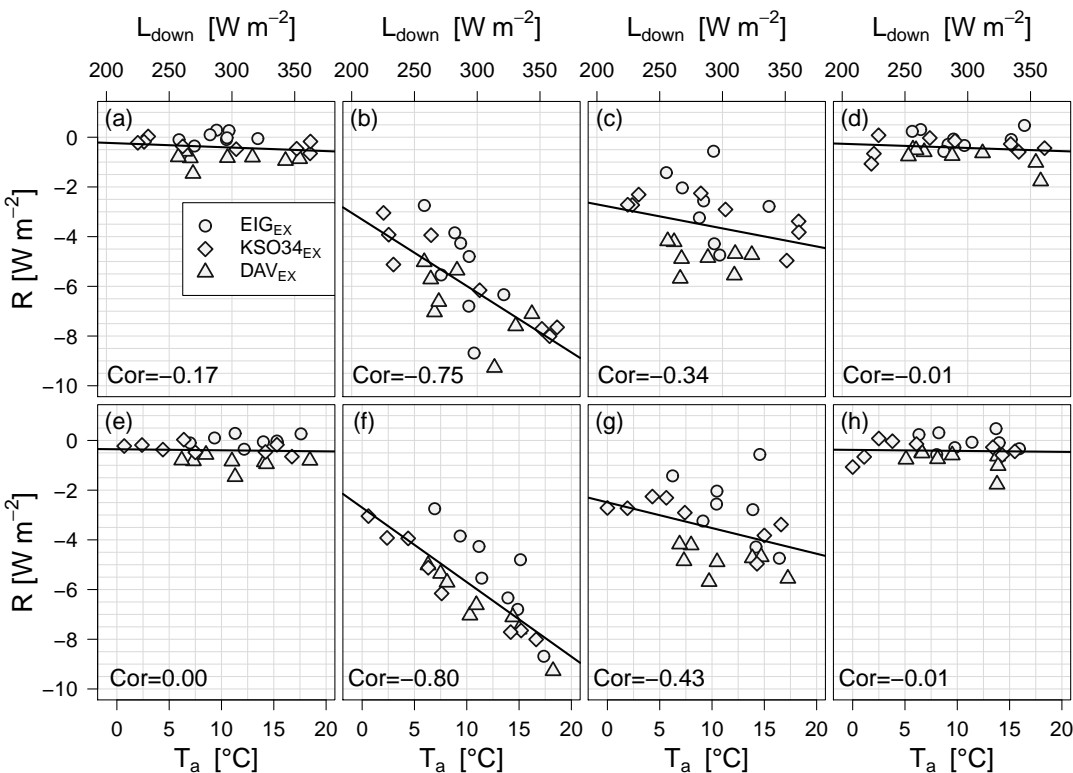

**Figure 6.** Minimum radiation measurements (maximum pyranometer responses) during nighttime as function of (a)-(d) longwave downward radiation ($L_{down}$, from ARAD site Graz/University) and (e)-(h) ambient air temperature ($T_a$, measured by LUFFT) during the field campaign in April/May 2016. Individual panels show minimum values of R and $L_{down}$ for selected time intervals: (a) $15-5$ min before spray-tests, (b) within $0-5$ min after spray-tests, (c) $15-25$ min after spray-tests, (d) $105-115$ min after spray-tests. (e)-(h) as (a)-(d) but for comparison with $T_a$. The Spearman rank correlation coefficient (Cor) is provided in each panel.

among pyranometer/HV-system combinations between 70 min ($KSO34_{EX}$) and 105 min ($EIG_{EX}$).

The results from these laboratory experiments confirm the substantial influence of precipitation events on pyranometer measurements as observed during routine observation in the ARAD network. Furthermore significant zero offsets occur independently of the HV-system used and recovery to initial state exceeds 1 h throughout. These results motivated a series of experiments under ambient environmental conditions described below, directed towards a better understanding of the magnitude of pyranometer offsets due to precipitation events.

## 3.2 Field experiments

Following the laboratory experiments two extensive field campaigns have been performed in January and April/May 2016. During these campaigns parallel measurements with a series of CMP21 pyranometers have been performed in direct vicinity to ARAD site Graz/University. The measurement setup comprised one reference pyranometer (REF, contained in EIG) and three experimental pyranometers (EX, contained in EIG, KSO34 and DAV HV-systems, respectively). Additionally ambient meteorological conditions (air temperature,

air humidity, wind direction, wind speed) have been monitored with an ,all-in-one' meteorological observing system, WS600 UMB manufactured by Lufft GmbH, (hereinafter referred to as LUFFT).

As our main interest lies in studying pyranometer zero-offsets and the effect of precipitation events, the majority of experiments has been performed during nighttime conditions. When ambient environmental conditions allowed (no natural precipitation), three experiments have been performed per night with a 2.5 h dormant phase between individual experiments. The dormant phase has been reduced by 30 min compared to laboratory experiments following the initial result of pyranometer signal recovery to initial state after laboratory spray-tests. Naturally nighttime conditions are less relevant to radiation monitoring though they provide a natural reference framework for the determination of instrument offsets. Furthermore any type of lens effect due to drop formation following precipitation events can be ruled out during nighttime conditions. Because of the potential relevance for estimating the effect of precipitation events on radiation monitoring accuracy during routine daytime operation, an additional series of spray-tests was performed under daylight conditions.

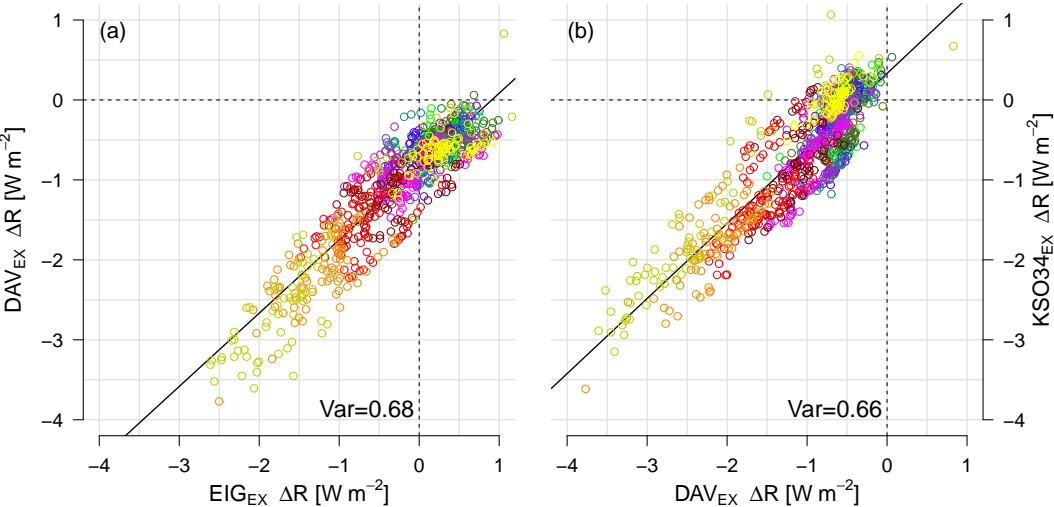

**Figure 7.** Scatter plots of the difference ($\Delta R$) between radiation output of individual experimental pyranometer/HV-system combinations ($\text{EIG}_\text{EX}$, $\text{KSO34}_\text{EX}$ and $\text{DAV}_\text{EX}$) during nighttime. In both panels the pyranometer sprayed second is given as a function of the one sprayed first. The colours mark bins (size 10 min with 1 min resolution) of measurements ranging from 10 min before (yellow) to 120 min after (brown) the spray-tests (see color bar in Fig. 5a for temporal evolution of experiments). The relationship between two pyranometers is characterized by the squared Spearman rang correlation coefficient Var.

### 3.2.1 Comparison of pyranometers under undisturbed conditions

First we turn the focus to the comparison of pyranometers contained in different HV-systems under ambient, undisturbed, nighttime conditions. Figure 3c summarizes results for both field campaigns. Comparison with laboratory experiments (Fig. 3a and b) show that the range of pyranometer output increases during ambient conditions reaching values of up to $2.4 \ \text{W m}^{-2}$. Nevertheless, the median difference in radiation measurements $\widetilde{\Delta R}$ between laboratory (Fig. 3b) and field (Fig. 3c) conditions yields very similar results for individual pyranometer/HV-system pairs: $|\widetilde{\Delta R}_\text{EIG}| = 1.35 \ \text{W m}^{-2}$, $|\widetilde{\Delta R}_\text{KSO34}| = 1.30 \ \text{W m}^{-2}$ and $|\widetilde{\Delta R}_\text{DAV}| = 1.37 \ \text{W m}^{-2}$.

### 3.2.2 Spray-tests during nighttime

Following the initial state comparison, a series of spray-tests has been performed for each pyranometer/HV-system combination under ambient environmental conditions. The automated spray-tests were performed for one system at a time, i.e. while one system was sprayed the reference system and all other experimental systems remained undisturbed. Figure 4b shows a time series of pyranometer measurements during spray-tests performed with a CMP21 pyranometer contained in a KSO34 HV-system. The output signal of $\text{KSO34}_\text{EX}$ decreased by $4 \ \text{W m}^{-2}$ (or more) during each experiment, a result very similar to experiments under laboratory conditions (Fig. 4a). This holds true also for other pyranometer/HV-system combinations, see Fig. S1b ($\text{EIG}_\text{EX}$) and Fig. S2b ($\text{DAV}_\text{EX}$) in the supplemental material.

The response of pyranometer/HV-system combinations to spray-tests under ambient environmental conditions is further explored in subpanels (d)-(f) (first field campaign, January 2016) and in subpanels (g)-(i) (second field campaign, April/May 2016) of Fig. 5. Here pairwise differences between experimental systems and REF during spray-tests under ambient environmental conditions are shown. Each comparison combines measurements of eight spray-tests, starting 10 min before and ending 120 min after each spray-test (marked with zero).

Independent of the HV-system used, all pyranometers reacted immediately to spray-tests and reached their maximum response (minimum value) within 5 min after the test. Overall responses are, in magnitude, similar among evaluated systems, and comparable to laboratory results.

Nevertheless, the time period needed by pyranometers/HV-systems to recover to their initial states varied among the instruments. Under laboratory conditions the average time needed to recover is similar for $\text{EIG}_\text{EX}$ and $\text{DAV}_\text{EX}$, while pyranometers contained in $\text{KSO34}_\text{EX}$ recover significantly faster. Under ambient environmental conditions the recovery times of $\text{EIG}_\text{EX}$ and $\text{DAV}_\text{EX}$ are 55 and 58 min respectively, which are approximately half their respective laboratory values ($\text{EIG}_\text{EX}$: 105 min and $\text{DAV}_\text{EX}$: 100 min), while for $\text{KSO34}_\text{EX}$ recovery times are not significantly different during ambient and laboratory conditions. Overall the results indicate a faster recovery of the pyranometer response under ambient environmental conditions (attributed

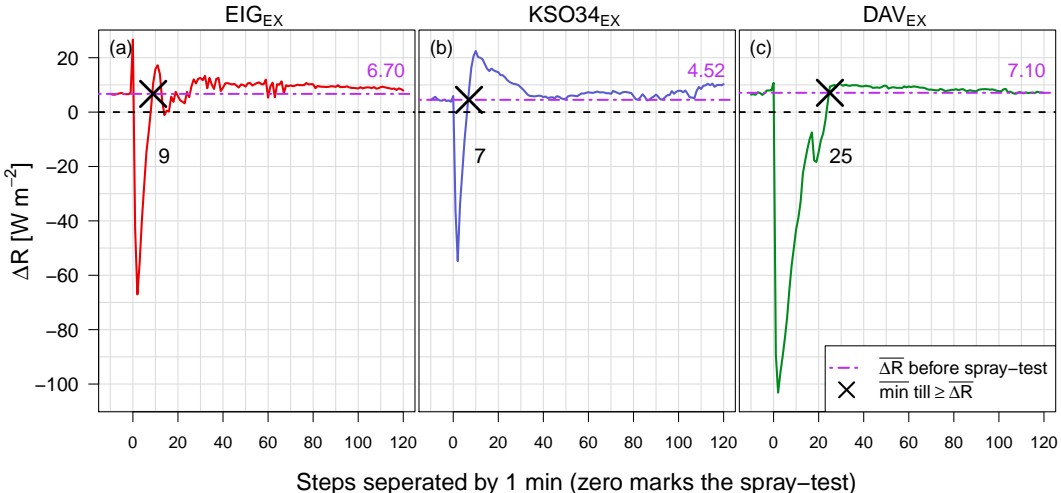

**Figure 8.** Difference in radiation ($\Delta R$) between the individual pyranometer/HV-system combinations and the reference pyranometer (REF) with a 1 min resolution during daytime: (a) EIG$_{EX}$, (b) KSO34$_{EX}$ and (c) DAV$_{EX}$. The x-axis shows time, starting $10$ min before and ending $120$ min after the spray-tests (marked with zero). The purple dotdashed horizontal line marks the average difference $\overline{\Delta R}$ before spray-test, the black cross marks the average time in minutes which each pyranometer/HV-system combination needed to return to/exceed its initial state. Note, these experiments have not been performed in parallel for technical reasons.

mainly to enhanced drying due to wind and ambient air temperature), in contrast to laboratory conditions. Nevertheless, all pyranometers required substantial time (at least $53$ min) to return to their initial signal states after the simulated precipitation events.

An interesting aspect is the relationship between pyranometer response and ambient air temperature $T_a$, as air temperature increases the variance in pyranometer response to spray-tests. The influence of $T_a$ is directly linked to longwave downward radiation ($L_{down}$), which governs $T_a$. While during undisturbed conditions a moderate relationship between pyranometer offsets and $L_{down}$ and $T_a$ is found, precipitation largely overwhelms infrared effects. Figure 6 illustrates this almost linear relationship, i.e. the larger $L_{down}$/higher $T_a$, the larger the pyranometer response following a (simulated) precipitation event, for results of the April/May field campaign. Panels (a) and (e) show scatterplots of $L_{down}$ and R and $T_a$ and R, respectively for the time interval $15-5$ min prior to spray-tests. Conversely panels (d) and (h) show the same relationships for the interval $105-115$ min following spray-tests. Immediately following spray-tests a significantly larger system response emerges leading also to a stronger connection with $L_{down}$ and $T_a$ (see panels (b) and (f)), which reduces over time as the system recovers towards its initial state (see panels (c) and (g) for the time interval $15-25$ min after spray-tests). The relationship between pyranometer response and ambient air temperature becomes also visible when comparing results of the April/May (Fig. 5g-i) and January field campaigns (Fig. 5d-f).

Since the standard setup for spray-tests during the field campaigns did not allow for a one-to-one comparison of pyranometer responses during spray-tests, an additional set of experiments was performed where individual experimental pyranometers have been sprayed in series under the same ambient environmental conditions. Figure 7 shows a one-to-one comparison of consecutively sprayed instruments, i.e., DAV$_{EX}$ as a function of EIG$_{EX}$ and KSO34$_{EX}$ as a function of DAV$_{EX}$. The colours mark bins (size 10 min) of measurements ranging from 10 min before (yellow) to 120 min after (brown) the spray-tests (see color bar in Fig. 5a for temporal evolution of experiments). The results show a good agreement among individual system responses, with an explained variance (squared Spearman rang correlation coefficient) of Var $= 0.68$ between EIG$_{EX}$ and DAV$_{EX}$ and Var $= 0.66$ between DAV$_{EX}$ and KSO34$_{EX}$.

### 3.2.3 Spray-tests during daytime

Having established pyranometer responses to simulated precipitation events, our focus shifted to the analysis of instrument responses under ambient daytime conditions. To this aim a series of spray-tests has been performed for each pyranometer/HV-system pair at the end of the second field campaign. The subpanels in Fig. 8 illustrate one test each for pyranometers contained in the three considered HV-systems. As expected, pyranometer responses are larger during daytime conditions reaching differences to REF of up to $-100$ W m$^{-2}$. The time needed for the sensors to recover to the initial states was significantly shorter than during nighttime conditions (EIG$_{EX}$: 9 min, KSO34$_{EX}$: 7 min and DAV$_{EX}$: 25 min), indicating rapid sensor adjustment. The larger system response but shorter recovery time indicates that recovery depends strongly on evaporation, i.e., the stronger the evaporation of the droplets on the glass-dome, due to ambient tem-

perature and wind speed, the smaller the time a pyranometer needs to recover to its initial state. A reasonable explanation considering evaporation depends on the radiation budget, temperature, relative humidity, and wind (Kraus, 2004).

## 4   Discussion and conclusions

This study seeks to investigate the influence of instrumentation and precipitation events on the accuracy of radiation measurements within the Austrian RADiation monitoring network (ARAD). To this aim one laboratory and two field campaigns have been performed in 2016, investigating pyranometer performance in different heating and ventilation systems (HV-systems) as well as zero offsets triggered by precipitation events. During the campaigns pyranometers of type CMP21 (Kipp&Zonen) have been operated as ‚experimental' in three different HV-systems (Eigenbrodt SBL 480 $EIG_{EX}$, DAVOS-PMOD/WRC $DAV_{EX}$ and the self-built $KSO34_{EX}$) and measurements have been compared with output of an undisturbed reference pyranometer (CMP21, contained in a housing of type Eigenbrodt SBL 480). To determine the effect of precipitation on measurement accuracy a series of more than 115 simulated precipitation events (as standardized spray-tests) has been performed.

The first campaign (January 2016) comprised a series of laboratory experiments at Kanzelhöhe Observatory. Results of the campaign showed that: (i) under undisturbed dormant conditions pyranometer output values lie within $\pm 1\,\mathrm{W\,m^{-2}}$, independent of the HV-system; (ii) standardized spray-tests (3.4 ml of distilled water) trigger zero offsets of $-4\,\mathrm{W\,m^{-2}}$ or more; (iii) the time individual pyranometer/HV-system combinations needed to recover to initial states after spray-tests differed but exceeded for all systems $70\,\mathrm{min}$. The pronounced pyranometer response following spray-tests is attributed to a change in the thermal balance, i.e., the evaporative cooling of the outer glass dome.

Following the laboratory campaign two intensive field campaigns have been performed in January and April/May 2016 in direct vicinity of ARAD site Graz/University at the measurement platform of the University of Graz. During field campaigns the same setup for radiation measurements (three experimental and one reference pyranometer) was used as during the laboratory campaign. Additionally ambient meteorological conditions (air temperature, air humidity, precipitation, wind speed and direction) have been monitored with an ‚all-in-one' meteorological observing system (LUFFT).

Results of the field campaign showed that: (i) the range of pyranometer output increases during ambient nighttime conditions reaching values of up to $2.4\,\mathrm{W\,m^{-2}}$; (ii) all pyranometers reacted immediately to spray-tests and reached their maximum response (minimum value) within $5\,\mathrm{min}$ of the test; (iii) pyranometer responses are similar among evaluated systems and comparable to laboratory results; (iv) individual pyranometer/HV-system combinations recovered

faster to their initial states following spray-tests under ambient environmental conditions, which is mainly attributed to enhanced drying due to wind and ambient air temperature. Further a quasi-linear relationship between the strength of the pyranometer response (decrease) after a spray-test and longwave downward radiation/ambient air temperature was found.

An additional set of experiments performed during daylight conditions indicates a significant effect of precipitation events during routine radiation monitoring. Differences to the undisturbed reference system reached up to $-100\,\mathrm{W\,m^{-2}}$ and sensors recovered substantially faster (within a few minutes) to initial states than during nighttime conditions, which is attributed to evaporation effects.

In summary the results from the series of laboratory and field experiments show a stable and comparable performance of CMP21 pyranometers throughout the different HV-systems used within the ARAD network. A significant effect of precipitation on the accuracy of daytime radiation measurements and nighttime zero offsets was found independent of the pyranometer/HV-system combination. The substantial time individual systems need to recover to stable initial states after precipitation events motivates flagging recommendations for operational use in the ARAD network. Precipitation data are available at all ARAD sites from co-located meteorological stations. We recommend flagging radiation measurements during/after precipitation events as system stability is not ensured as our results show. Our recommendations are: (i) flagging daytime radiation measurements as ‚wrong' during precipitation events and ‚dubious' for 10 minutes following precipitation events; (ii) flagging of nighttime outputs as ‚wrong' during precipitation events and ‚dubious' for 90 minutes following precipitation events. Furthermore, we recommend applying the same flagging criteria/intervals as for precipitation events for routine pyranometer cleanings, if water or alcohol is sprayed on the pyranometer's outer glass dome. Similar flagging criteria might be useful to improve meta-data information also in other radiation monitoring networks.

We note in closing, that additional field and/or laboratory experiments characterizing pyranometer offsets following abrupt temperature changes and for different precipitation types (e.g., snow, freezing rain, rain and snow mixes) would strongly increase our understanding of the influence of ambient meteorology, and abrupt changes therein, on the stability and measurement accuracy of BSRN class pyranometers in different HV-systems. Further additional analysis regarding offsets following precipitation events for unventilated pyranometers are recommended.

*Acknowledgements.* The authors thank the Austrian Central Institute for Meteorology and Geodynamics (ZAMG) for providing CMP21 pyranometers and Eigenbrodt SBL 480 and DAVOS-PMOD/WRC HV-systems operated during campaigns, as well as data from TAWES site Graz/University and ARAD site

Graz/University. The authors are grateful to Martin Mair (ZAMG) for technical support and fruitful discussions, Erich Mursch-Radlgruber (University of Natural Resources and Applied Sciences (BOKU) Vienna) for providing an ‚all-in-one' meteorological observing system (WS600 UMB by Lufft GmbH), and Heinrich Freislich (Kanzelhöhe Observatory) for technical support and constructions. S. Oswald acknowledges financial support through fellowships from the Department of Environmental, Regional and Educational Sciences of the University of Graz and Grazer Wechselseitige Versicherungen AG. The authors thank Joseph Michalsky and an anonymous referee for their helpful comments during the discussion phase of this article.

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
