# Peer review of "Pyranometer offsets triggered by ambient meteorology: insights from laboratory and field experiments"

_Atmospheric Measurement Techniques, 2016_

## Referee Comment (RC1) · J. Michalsky (Referee) · 1 Dec 2016

The paper examines the effects of liquid precipitation on pyranometer output under laboratory and ambient conditions (commonly referred to as offsets). It also looks at the effects of three different ventilation systems for the same pyranometer type. I find the experiments were carefully conducted and add new information that should allow one to scrutinized irradiance data with an eye toward eliminating unphysical results after precipitation events. To my knowledge this type of study has not been performed, but was needed to explain strange results that were suspected, but until now, not confirmed by experiments.

I would add a comment to the text that this affects data taken right after routine pyra-

nometer cleaning when water or alcohol is sprayed on the pyranometer's outer glass.

It follow on experiments, it would be interesting to see how snow, wind, and rapid temperature changes affect offsets.

---

## Author Comment (AC1) · 15 Dec 2016

We thank Dr. Michalsky (Referee 1) for the positive judgment of our work and his suggestions for revising the manuscript.

Below we provide the review (in bold) and our point to point response to individual comments.

**The paper examines the effects of liquid precipitation on pyranometer output under laboratory and ambient conditions (commonly referred to as offsets). It also looks at the effects of three different ventilation systems for the same pyranometer type. I find the experiments were carefully conducted and add new**

**information that should allow one to scrutinized irradiance data with an eye toward eliminating unphysical results after precipitation events. To my knowledge this type of study has not been performed, but was needed to explain strange results that were suspected, but until now, not confirmed by experiments.**

We thank the referee for acknowledging the originality of our work and its contribution towards eliminating unphysical results in radiation measurements.

**I would add a comment to the text that this affects data taken right after routine pyranometer cleaning when water or alcohol is sprayed on the pyranometer's outer glass.**

This is an excellent point. We will include a statement discussing effects on data reliability after pyranometer cleaning (with water or alcohol) in the discussion section of the revised manuscript.

**It follow on experiments, it would be interesting to see how snow, wind, and rapid temperature changes affect offsets.**

From our set of field-experiments we have a small set of spray-tests (during the January field campaign) at temperatures below 0 °C available. These spray-tests led to 'freezing rain' on the pyranometer glass dome. While the initial sensor response to 'freezing rain' was similar as observed during 'liquid precipitation' it took the sensor longer to recover to initial state. The small set of spray-tests below 0 °C available however does not allow drawing statistically robust conclusions and these results are therefore not included in the present manuscript. We agree that follow up experiments characterizing pyranometer offsets following abrupt temperature changes and different precipitation types would be highly interesting. Such experiments would require a more comprehensive laboratory equipment (e.g., a climate chamber) and could be performed, possibly with an extension towards other pyranometer types and heating/ventilation systems, in a community effort. We will include a statement indicating potential future directions in the discussion section of the revised manuscript.

---

## Referee Comment (RC2) · Anonymous Referee #2 · 13 Feb 2017

In the manuscript "Pyranometer offsets triggered by ambient meteorology: insights from laboratory and field measurements" Oswald et al. discuss impact of the precipitation on the shortwave radiation measured by standard pyranometers with different ventilation systems. The conclusion from this study is very important and useful for radiation community. Recommended by authors flagging radiation during and after precipitation day and nighttime measurements should be applied by WMO, BSRN network. The manuscript is generally well written and clearly presented and therefore in my opinion this manuscript can be published in AMT after minor revision

Main concerns:

1. The main concern is lack of the information about response of the non-ventilation

pyranometers on the precipitation. Could you provide any results or some estimation of the impact? If not please provide some discussion about this kind of the radiometers. 2. Some information on the spray system is needed in the section 2. For example about droplet size which may important for radiometer response 3. Could add information on relative humidity during laboratory experiments?

---

## Author Comment (AC2) · 24 Feb 2017

We thank Referee 2 for the positive judgment of our work and the useful suggestions for revising the manuscript.

Below we provide the review (in bold) and our point to point response to individual comments.

**In the manuscript 'Pyranometer offsets triggered by ambient meteorology: insights from laboratory and field measurements' Oswald et al. discuss impact of the precipitation on the shortwave radiation measured by standard pyranometers with different ventilation systems. The conclusion from this study is very**

[Figure]

**important and useful for radiation community. Recommended by authors flagging radiation during and after precipitation day and nighttime measurements should be applied by WMO, BSRN network. The manuscript is generally well written and clearly presented and therefore in my opinion this manuscript can be published in AMT after minor revision.**

We thank the referee for acknowledging the originality of our work and its importance for the radiation community.

**The main concern is lack of the information about response of the non-ventilation pyranometers on the precipitation. Could you provide any results or some estimation of the impact? If not please provide some discussion about this kind of the radiometers.**

Our study was focusing on BSRN-class pyranometers (and HV-systems) operated in the Austrian RADiation network. As ARAD adopts BSRN recommendation regarding pyranometer operation in ventilated housings (e.g. McArthur, 2005) we have not investigated precipitation effects on non-ventilated pyranometers within the present study. Nevertheless, we agree with the referee that such analysis would be of immediate interest for the radiation community. We will include a statement regarding the importance of a similar analysis for non-ventilated systems in the discussion section of the revised manuscript.

**Some information on the spray system is needed in the section 2. For example about droplet size which may important for radiometer response 3.**

The spray system created very fine, homogeneous drizzle, producing small droplets on the pyranometer dome, which quickly coagulated to larger drops (see Fig. 1). Such coagulation on pyranometer domes is also observed during stratiform and convective precipitation events. We will provide this information in the revised manuscript.

**Could add information on relative humidity during laboratory experiments?**

Relative humidity has been $\approx 65\%$ throughout the series of laboratory experiments. We will include this information in the revised manuscript.

[Figure]

[Figure]

**Fig. 1.** Drizzle and coagulated drops on the pyranometer dome during a spray-test.

---

## Author Response (AR2)

We thank Dr. Michalsky (Referee 1) for the positive judgment of our work and his suggestions for revising the manuscript.

Below we provide the review (in bold) and our point to point response to individual comments.

The paper examines the effects of liquid precipitation on pyranometer output under laboratory and ambient conditions (commonly referred to as offsets). It also looks at the effects of three different ventilation systems for the same pyranometer type. I find the experiments were carefully conducted and add new

information that should allow one to scrutinized irradiance data with an eye toward eliminating unphysical results after precipitation events. To my knowledge this type of study has not been performed, but was needed to explain strange results that were suspected, but until now, not confirmed by experiments.

We thank the referee for acknowledging the originality of our work and its contribution towards eliminating unphysical results in radiation measurements.

I would add a comment to the text that this affects data taken right after routine pyranometer cleaning when water or alcohol is sprayed on the pyranometer's outer glass.

This is an excellent point. We will include a statement discussing effects on data reliability after pyranometer cleaning (with water or alcohol) in the discussion section of the revised manuscript.

**It follow on experiments, it would be interesting to see how snow, wind, and rapid temperature changes affect offsets.**

From our set of field-experiments we have a small set of spray-tests (during the January field campaign) at temperatures below 0 °C available. These spray-tests led to 'freezing rain' on the pyranometer glass dome. While the initial sensor response to 'freezing rain' was similar as observed during 'liquid precipitation' it took the sensor longer to recover to initial state. The small set of spray-tests below 0 °C available however does not allow drawing statistically robust conclusions and these results are therefore not included in the present manuscript. We agree that follow up experiments characterizing pyranometer offsets following abrupt temperature changes and different precipitation types would be highly interesting. Such experiments would require a more comprehensive laboratory equipment (e.g., a climate chamber) and could be performed, possibly with an extension towards other pyranometer types and heating/ventilation systems, in a community effort. We will include a statement indicating potential future directions in the discussion section of the revised manuscript.

СЗ

We thank Referee 2 for the positive judgment of our work and the useful suggestions for revising the manuscript.

Below we provide the review (in bold) and our point to point response to individual comments.

In the manuscript 'Pyranometer offsets triggered by ambient meteorology: insights from laboratory and field measurements' Oswald et al. discuss impact of the precipitation on the shortwave radiation measured by standard pyranometers with different ventilation systems. The conclusion from this study is very

important and useful for radiation community. Recommended by authors flagging radiation during and after precipitation day and nighttime measurements should be applied by WMO, BSRN network. The manuscript is generally well written and clearly presented and therefore in my opinion this manuscript can be published in AMT after minor revision.

We thank the referee for acknowledging the originality of our work and its importance for the radiation community.

The main concern is lack of the information about response of the nonventilation pyranometers on the precipitation. Could you provide any results or some estimation of the impact? If not please provide some discussion about this kind of the radiometers.

Our study was focusing on BSRN-class pyranometers (and HV-systems) operated in the Austrian RADiation network. As ARAD adopts BSRN recommendation regarding pyranometer operation in ventilated housings (e.g. McArthur, 2005) we have not investigated precipitation effects on non-ventilated pyranometers within the present study. Nevertheless, we agree with the referee that such analysis would be of immediate interest for the radiation community. We will include a statement regarding the importance of a similar analysis for non-ventilated systems in the discussion section of the revised manuscript.

**Some information on the spray system is needed in the section 2. For example about droplet size which may important for radiometer response 3.**

The spray system created very fine, homogeneous drizzle, producing small droplets on the pyranometer dome, which quickly coagulated to larger drops (see Fig. 1). Such coagulation on pyranometer domes is also observed during stratiform and convective precipitation events. We will provide this information in the revised manuscript.

Could add information on relative humidity during laboratory experiments?

Relative humidity has been  $\approx 65\%$  throughout the series of laboratory experiments. We will include this information in the revised manuscript.

СЗ

Fig. 1. Drizzle and coagulated drops on the pyranometer dome during a spray-test.

**Pyranometer offsets triggered by ambient meteorology: insights from laboratory and field experiments**

Sandro M. Oswald1, 2, 3, Helga Pietsch2, Dietmar J. Baumgartner4, Philipp Weihs3, and Harald E. Rieder1, 2, 5

1Wegener Center for Climate and Global Change, Graz, Austria

2Institute for Geophysics, Astrophysics and Meteorology/Institute of Physics, Graz, Austria

3Institute of Meteorology, University of Natural Resources and Applied Sciences (BOKU), Vienna, Austria

4Kanzelhöhe Observatory for Solar and Environmental Research, Graz, Austria

5Austrian Polar Research Institute, Vienna, Austria

Correspondence to: Sandro M. Oswald (sandro.oswald@boku.ac.at)

Abstract. This study investigates effects of ambient meteorology on the accuracy of radiation measurements performed with pyranometers contained in various heating/ventilation systems (HV-systems). It focuses particularly on instrument

- offsets observed following precipitation events. To quantify pyranometer responses to precipitation, a series of controlled laboratory experiments as well as two targeted field campaigns were performed in 2016. The results indicate that precipitation (as simulated by spray-tests or observed under am-
- 10 bient conditions) significantly affects the thermal environment of the instruments and thus their stability. Statistical analysis of laboratory experiments showed that precipitation triggers zero offsets of  $-4 \text{ Wm}^{-2}$  or more, independent of the HV-system. Similar offsets have been observed in field
- experiments under ambient environmental conditions, indicating a clear exceedance of BSRN targets following precipitation events. All pyranometers required substantial time to return to their initial signal states after the simulated precipitation events. Therefore for BSRN class measurements
- 20 the recommendation would be to flag the radiation measurements during a natural precipitation event and 90 min after it in nighttime conditions. Further daytime experiments show pyranometer offsets of 50 W m-2 or more in comparison to the reference system. As they show a substantially faster re-
- 25 covery, the recommendation would be to flag the radiation measurements within a natural precipitation event and 10 min after it in daytime conditions.

**1 Introduction**

Earth's climate is largely determined by the global energy balance (Wild et al., 2012). Therefore a precise knowledge of the surface energy budget, which includes the solar and terrestrial radiation fluxes, is essential for understanding the Earth's planetary circulation and climate system (Ramanathan, 1987; Augustine and Dutton, 2013; Wild et al., 2014).

In situ measurements of solar radiation on the Earth's surface, more precisely global radiation which is the sum of the direct and diffuse components, began in the 1920s, but became more widespread with the advent of thermopile pyranometers and through initiatives of the International Geophysical Year 1957/58 (Wild, 2009). Around the turn of the century a series of studies (Dutton et al., 1991; Gilgen et al., 1998; Ohmura et al., 1998; Stanhill, 2005; Liepert, 2002) reported negative trends of global radiation based on in-situ measurements, a phenomenon commonly referred to as ,global dimming' (Wild, 2005, 2009). Average trends of -6 to -9 W m-2 between 1960-1990 have been reported in the literature (Wild, 2005), but estimates vary depending on location, record length and time period considered (Wild et al., 2012). The previously observed negative trends have been replaced by a widespread increase in surface solar radiation over the period 1990-2000, a phenomenon commonly referred to as ,global brightening' (Wild, 2005).

The growing interest of the scientific community in surface radiation trends and limitations in the accuracy of historic records led in the early 1990s to the establishment of the Baseline Surface Radiation Network (BSRN) under the auspices of the World Climate Research Programme (WCRP) 40

45

30

50